# Visualizing a single wavefront dislocation induced by orbital angular momentum in graphene

Yi-Wen Liu [1,2,5], Yu-Chen Zhuang [3,5], Ya-Ning Ren[1,2,5], Chao Yan[1,2], Xiao-Feng Zhou[1,2], Qian Yang[1,2], Qing-Feng Sun [3,4] ✉ & Lin He [1,2] ✉

Phase singularities are phase-indeterminate points where wave amplitudes are zero, which manifest as phase vertices or wavefront dislocations. In the realm of optical and electron beams, the phase singularity has been extensively explored, demonstrating a profound connection to orbital angular momentum. Direct local imaging of the impact of orbital angular momentum on phase singularities at the nanoscale, however, remains challenging. Here, we study the role of orbital angular momentum in phase singularities in graphene, particularly at the atomic level, through scanning tunneling microscopy and spectroscopy. Our experiments demonstrate that the scatterings between different orbital angular momentum states, which are induced by local rotational symmetry-breaking potentials, can generate additional phase singularities, and result in robust single-wavefront dislocations in real space. Our results pave the way for exploring the effects of orbital degree of freedom on quantum phases in quasiparticle interference processes.

The phase of waves constitutes a fundamental parameter for all waves across diverse domains of physics[1–9]. Phase singularity is an essential aspect of phase, denoting a phase indetermination point where the wave amplitude becomes zero. Encircling a phase singularity results in the accumulation of a certain phase, thereby initiating the emergence of additional wavefronts, i.e., additional surfaces of constant phase or wavefront dislocations in the wave field[10–16]. In many cases, phase singularity is intimately related to orbital angular momentum. For instance, orbital angular momentum-carrying optical or electron beams usually exhibit spiral wavefronts, which involve phase singularities and further induce wavefront dislocations[17–22]. These phenomena are widely observed through interference experiments on both macroscopic and microscopic scales, which have been extensively utilized in numerous applications[17–27].

However, it remains a challenge to explore the effects of orbital angular momentum on phase singularities at the nanoscale in condensed matter physics, owing to the limited available methods. Here, in two-dimensional (2D) massless Dirac fermions, we decipher the significance of orbital angular momentum effects by directly visualizing wavefront dislocations in real space[9]. Specifically, by breaking the local rotational symmetry, inter-orbital angular momentum scatterings could occur, resulting in a transition of the interference pattern from two wavefront dislocations to a single one. The orbital angular momentum induced single-wavefront dislocations are universally observed in rotationally asymmetric systems, regardless of whether the anisotropy originates from the potential of the tip or the geometry of the sample.

## Results and discussion

### Inter-orbital angular momentum scatterings

Firstly, we briefly demonstrate the emergence of two wavefront dislocations in Friedel oscillations contributed by intervalley scatterings without inter-orbital angular momentum scatterings, a phenomenon also studied in previous experiments[9]. A process of intervalley

[1]Center for Advanced Quantum Studies, Department of Physics, Beijing Normal University, 100875 Beijing, China. [2]Key Laboratory of Multiscale Spin Physics, Ministry of Education, 100875 Beijing, China. [3]International Center for Quantum Materials, School of Physics, Peking University, 100871 Beijing, China. [4]Hefei National Laboratory, Hefei 230088, China. [5]These authors contributed equally: Yi-Wen Liu, Yu-Chen Zhuang, Ya-Ning Ren ✉e-mail: sunqf@pku.edu.cn; helin@bnu.edu.cn

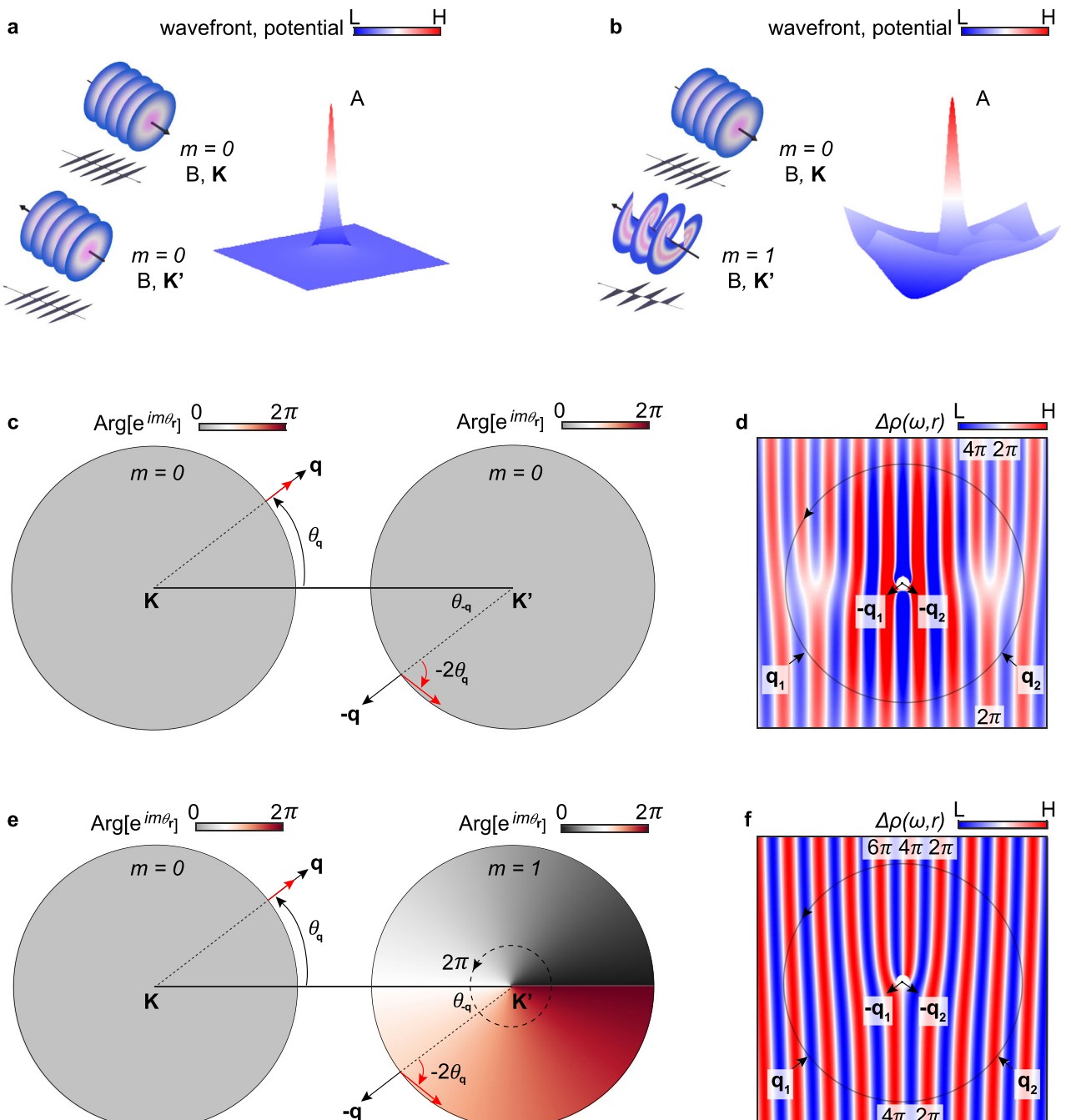

**Fig. 1 | Dynamics of intervalley scatterings and wavefront dislocations. a** A schematic diagram of intervalley interference between the incident wave from position B in **K** valley and the reflected wave back to position B in **K'** valley due to the defect potential A. The orbital angular momentum $m$ is zero for these wavefronts. The color scale represents the intensity distributions for both wavefronts and the potential. **b** A wave scattered by a defect potential accompanied by a spread rotationally asymmetric potential, leading to reflection as a spiral wavefront with a finite orbital angular momentum. **c** A intervalley scattering process for panel **a** between two nearest **K** and **K'** valleys, involving the pseudospin rotation. Here $\theta_{\mathbf{r}}$ is the polar angle of the position vector **r**. Black and red arrows denote the vector of momentum **q** and pseudospin, respectively. $\theta_{\mathbf{q}}$ is rotation angle of momentum. The color gradient from black to red represents the variation of the phase. **d** A schematic diagram to show

wavefront dislocations in total local density of states(LDOS) with the contribution of pseudospin rotation. Panels **a** and **c** are the partial processes of panel **d**. The **q** in all the directions scatters with the potential in the middle point and contributes to the accumulation of the phase for the closed loop. Two scattering directions $\mathbf{q_1}$ and $\mathbf{q_2}$ are shown in the figure. The phases from $2\pi$ to $4\pi$ are marked in one of the wavefront dislocations and the value of the wave is donated by different colors. **e** A intervalley scattering process between two nearest **K** and **K'** valleys for panel **b**, involving both pseudospin and orbital angular momentum. The orbital angular momentum effect on valley manifests as the colored distribution with a varied phase. **f** The wavefront dislocation in LDOS modulation $\Delta\rho_B(\Delta\mathbf{K},\mathbf{r}) \propto \cos(\Delta\mathbf{K}\cdot\mathbf{r}+\theta_{\mathbf{r}})$ with the contribution from both pseudospin rotation and orbital angular momentum coupling. The white dots in **d** and **f** schematically denote the defect position.

scattering induced by the atomic defect or adatom in graphene is schematically shown in Fig. 1a. An incident wave from sublattice B in **K** valley, is scattered by the defect-induced local potential located at sublattice A, and then propagates back to the original position in **K'**

valley. Since the local potential behaves like an isotropic hard wall, the intervalley scattering process hardly induce a change in the orbital angular momentum of electrons. Instead, due to the distinctive pseudospin textures for two valleys as shown in Fig. 1c, the intervalley

scattering along intervalley scattering direction $\Delta\mathbf{K} = \mathbf{K} - \mathbf{K}'$ causes pseudospin vector in sublattice space rotating by an angle of $\theta_{pseudo} = -2\theta_{\mathbf{q}}$ (here $\theta_{\mathbf{q}}$ is the polar angle of the incident wave with momentum $\mathbf{q}$), which comes from the two-component wave function, and does not occur in scattering processes of the conventional two-dimensional electron gases (2DEG)[9,28–32]. This pseudospin rotation related momentum $\mathbf{q}$ will further affect the spatial interference pattern and induce wavefront dislocations in Friedel oscillations. To see this, we can do Fourier-filtering by only picking out the elastic backscattering along $\pm\Delta\mathbf{K}$ corresponding to two peaks in Fast Fourier Transform (FFT), which are 30 degree rotated reciprocal lattice vectors relative to graphene[33,34]. By selecting peaks at the specific frequency in FFT, we can obtain the local density of states (LDOS) modulation $\Delta\rho$ at this frequency through the inverse FFT. This LDOS modulation $\Delta\rho$ results from the quantum interference between the incident and the scattering wave, corresponding to $\lambda = \frac{2\pi}{|\Delta\mathbf{K}|} \approx 3.7\text{Å}$ periodic cosine wave in real space, which is shown in Fig. 1a, d (the color is the amplitude of the wave, and each stripe reflects the phase of the cosine wave). Specially, the pseudospin rotation as shown in Fig. 1c will contributes an additional phase shift $2\theta_{\mathbf{r}}$ into LDOS fluctuations as a topological phase singularity: $\Delta\rho_B(\Delta\mathbf{K},\mathbf{r}) \propto \cos(\Delta\mathbf{K}\cdot\mathbf{r} + 2\theta_{\mathbf{r}})$ (see Supplementary Note 1 and refs. 9,30). Here $\theta_{\mathbf{r}}$ is the real space polar angle of $\mathbf{r}$ relative to the defect, and the phase $2\theta_{\mathbf{r}}$ becomes singular at $\mathbf{r} = 0$, and this singularity is also associated with the defect position in real space, which will be discussed later.

Since the $\Delta\rho_B$ is a single-valued function and must return to the same value after encircling the singularity along a closed loop, the corresponding phase accumulation should be a quantized topological number $2\pi N$ (N is an integer), which can generate additional wavefronts in LDOS. In real space, after encircling the defect (singularity) with the STM tip, $2\theta_{\mathbf{r}}$ accumulate $2*2\pi = 4\pi$ phase ($\theta_{\mathbf{r}}$ changes from 0 to $2\pi$ for a circle). Therefore, two wavefront dislocations can emerge in $\Delta\rho_B(\Delta\mathbf{K},\mathbf{r})$, as illustrated in Supplementary Fig. 1. Furthermore, we can also consider a similar intervalley scattering process for LDOS modulation on sublattice A, which does not exhibit any phase singularity: $\Delta\rho_A(\Delta\mathbf{K},\mathbf{r}) \propto \cos(\Delta\mathbf{K}\cdot\mathbf{r})$ (see Supplementary Note 1 and refs. 9,30). In Fig. 1d, we schematically show the total charge density modulation $\Delta\rho(\Delta\mathbf{K},\mathbf{r}) = \Delta\rho_A(\Delta\mathbf{K},\mathbf{r}) + \Delta\rho_B(\Delta\mathbf{K},\mathbf{r})$ (see details in Supplementary Note 1). Clearly, the result exhibits double Y-shaped fringes which is well consistent with previous study[9]. Additionally, as the momentum $\mathbf{q}$ circles the Dirac point along an anticlockwise-oriented contour, the pseudospin will rotate $2\pi\xi$ with $\xi = \pm1$ for $\mathbf{K}$ and $\mathbf{K}'$ valley, which yields the topological number (winding number) $W = \xi$ as well as the Berry phase $\gamma = \xi\pi$ (see Supplementary Note 1). Since the geometric phase shift $2\theta_{\mathbf{r}}$ in LDOS modulation is directly related to the pseudospin rotation angle $2\theta_{\mathbf{q}}$, $N = 2$ additional wavefronts following $2\pi N = 4|\gamma|$ also strongly reflects the topology of Dirac cones[9].

To investigate the role of orbital angular momentum in scattering processes and its impact on the interference in LDOS, we introduce an anisotropic potential that extends over a range surrounding the defect potential, exhibiting a directional dependence at the atomic scale. In this case, the conservation of orbital angular momentum in the scattering process is broken, leading to the occurrence of inter-orbital angular momentum scatterings. At this time, except for a relative phase between A and B sublattice from the pseudospin-locking texture shown in the Fig. 1c, an additional term of $e^{im\theta_{\mathbf{r}}}$ on both A and B sublattices related to orbital angular momentum $m$ could also contribute (see definitions in Supplementary Note 2).

As an example, in Fig. 1b, one incident wave ($\mathbf{K}$ valley) with $m = 0$ orbital angular momentum (similar to a plane wave) is scattered by local anisotropic potential on sublattice A. It then returns as a reflected wave ($\mathbf{K}'$ valley) to the original position but possesses an orbital angular momentum $m = 1$, forming like a spiral wave. In particular, the incident wave has a constant phase structure, while the reflected wave possesses a helical phase structure represented

by $e^{i\theta_{\mathbf{r}}}$, as shown by the color in Fig. 1e. Therefore, the interference between the incident wave and reflected wave leads to the additional induction of a phase $\theta_{\mathbf{r}}$ from orbital angular momentum difference. Now, both contributions from the pseudospin vector rotation and the interaction among orbital angular momentum states can shift the phase in LDOS modulation for B sublattice from $2\theta_{\mathbf{r}}$ to $2\theta_{\mathbf{r}} - \theta_{\mathbf{r}} = \theta_{\mathbf{r}}$ with $\Delta\rho_B(\Delta\mathbf{K},\mathbf{r}) \propto \cos(\Delta\mathbf{K}\cdot\mathbf{r} + \theta_{\mathbf{r}})$ (see Supplementary Note 2). As the tip encircles the defect, an accumulation of $2\pi$ phase occurs, resulting in a change from two wavefront dislocations into a single wavefront dislocation, as depicted in Fig. 1f.

## Imaging single-wavefront dislocations

To explore the above-mentioned inter-orbital angular momentum scattering effects on Friedel oscillations at the nanoscale, we performed a quasiparticle interference (QPI) experiment utilizing scanning tunneling microscopy (STM) and density-tuned scanning tunneling spectroscopy (STS). The device geometry is shown in the top panel of Fig. 2a. To obtain the hydrogen chemisorption defect-rich graphene samples[35], we either employ high-hydrogen growth graphene and subsequently transfer it onto hexagonal boron nitride (hBN) substrates[36,37], or apply tip pulses directly onto graphene/germanium (Ge) heterostructures (Supplementary Fig. 2). The lower panel of Fig. 2a illustrates the relationship between intervalley scattering processes in momentum space and the electron trajectory in real space in the STM experiment. Electrons with momentum $\mathbf{q}$ can be injected towards the defect by the STM tip at the position $\mathbf{r}$ relative to the defect. As shown in the panel a, the polar angle $\theta_{\mathbf{q}}$ is directly related to the polar angle $\theta_{\mathbf{r}}$ of the tip in real space by $\theta_{\mathbf{r}} = \theta_{\mathbf{q}} + \pi$. Since the $\theta_{\mathbf{r}}$ and $\theta_{\mathbf{q}}$ are closely intertwined, the rotation of pseudospin vectors $2\theta_{\mathbf{q}}$ in momentum space can naturally generate phase singularities $2\theta_{\mathbf{r}}$ in real space and wavefront dislocations in LDOS map through the relation $\theta_{\mathbf{r}} = \theta_{\mathbf{q}} + \pi$ (ref. 9), as shown in Figs. 1 and 2a.

In our experiment, the hydrogen chemisorption defect in graphene exhibits two distinct types of wavefront dislocations, as summarized in Fig. 2. Figures 2b, f shows two representative topographic images. Both defects have three-fold rotational symmetry with the highest site localized at the center of a carbon atom, exhibiting similar features as a single-atom absorption in graphene[9]. The atomic defects can generate strong intervalley scattering, characterized as a threefold $\sqrt{3}\times\sqrt{3}R30°$ scattering pattern with respect to the graphene lattice, as shown in the STM topography (Fig. 2b, f). The signals of the intervalley scattering can be clearly observed also in their FFT images[33,34], which show six inner bright spots rotated by 30 degrees with respect to the outer Bragg peaks (Fig. 2c, g).

As introduced in Fig. 1a, c, d, the pseudospin vector rotation $2\theta_{\mathbf{q}}$ induced by intervalley scattering leads to $2*2\pi$ phase accumulation when encircling the defect and, consequently, can generate two wavefront dislocations ($2\pi$ phase induces one additional equiphase line, which corresponds to one wavefront dislocation)[9,31,32]. To elucidate this point, we isolate the intervalley signals and employ an inverse FFT to acquire the wavefront dislocations in a specific direction in real space, as illustrated in Fig. 2d. The wavefront image contains two parts of information. One aspect concerns the amplitude of the wavefunction, and the other aspect relates to the phase. Two wavefront dislocations are observed around defect A due to the $4\pi$ phase accumulation, in line with previous studies[9,31,32]. The 3.7 Å-wavelength LDOS modulations originate from the momentum transfer vector at the iso-energy contour between different valleys[33,34], which is in agreement with the calculation incorporating only a defect-induced delta potential[9,31,32], as depicted in Fig. 2e (see Supplementary Notes 3, 4 for further details). It is worth noting that the appearance of two wavefront dislocations is consistently observed in various substrates, temperatures and under distinct filter conditions (Supplementary Fig. 2). All these two wavefront dislocations are localized in the vicinity of the defect within the distance of approximately 15 wavelengths

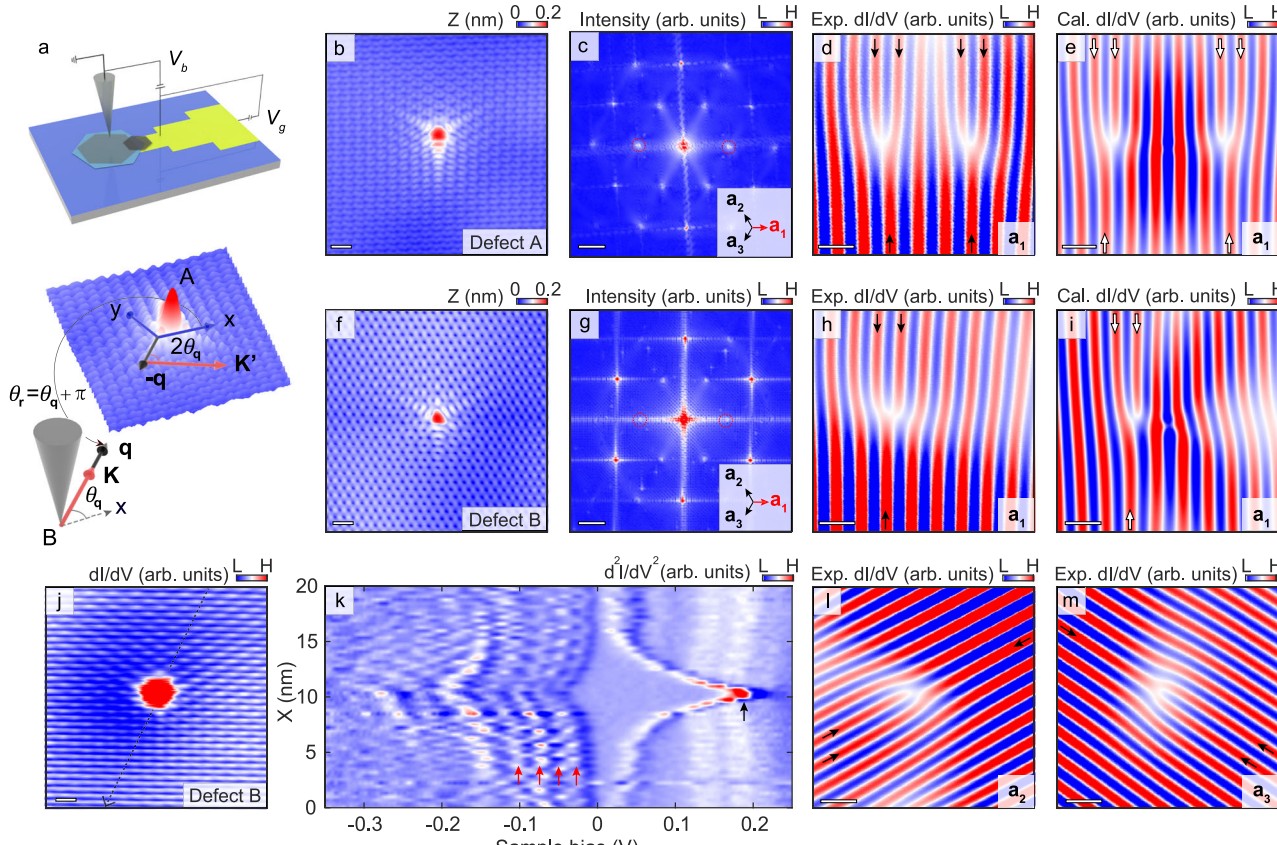

**Fig. 2 | Two types of wavefront dislocations in experiments. a** Experimental setup (graphene/hexagonal boron nitride (hBN)/silicon back-gate). $V_b$ and $V_g$ are the sample bias and back-gate voltage, respectively. The intervalley scattering process is detected by the scanning tunneling microscopy (STM) tip, occurring between two valleys, and the pseudospin vector rotates by an angle of -$2\theta_q$. The $\theta_r = \theta_q + \pi$ relation is marked in the figure. **b** STM topography of a hydrogen chemisorption defect detected by the tungsten tip (defect A, $V_b = 0.45$ V). **c** Fast Fourier Transform (FFT) image of the defect A, and six inner peaks come from the intervalley scattering process. Right lower inset, $\mathbf{a_1}$, $\mathbf{a_2}$, and $\mathbf{a_3}$ directions, which are defined according to the direction of the intervalley scattering in FFT, are labeled. Scale bar: 10 nm⁻¹. **d** Experimental wavefront dislocations, which is extracted by filtering the red circles in panel **c**. Both $\pm \Delta\mathbf{K}$ are considered during the filtering. The splitting of the wavefront dislocations is due to the contribution from both A and B

sublattices. **e** Theoretical LDOS modulation at the energy of about 0.44 V. **f** Topography for another hydrogen chemisorption defect obtained by Au-coated tip (defect B, $V_b = 0.2$ V). **g** FFT image of the defect in **f**. Scale bar: 10 nm⁻¹.
**h**, **i** Experimental and theoretical (the energy is about 0.26 eV) wavefront dislocations of defect B, showing one wavefront dislocation. **j** $dI/dV$ map obtained at $V_b = 0.2$ V, which shows asymmetric potential in real space. **k** Derivative STS point spectra collected along the black line direction in panel **j** with a longer distance of about 20 nm, showing quasibound states (red arrows) and a charging peak (black arrow). **l**, **m** Robust single-wavefront dislocations in the other two directions ($\mathbf{a_2}$ and $\mathbf{a_3}$). **f**–**m** are obtained at the same defect. The scale bars are 0.5 nm in all real-space figures. The changes of wavefronts passing through the defects are indicated by the dark (white) arrows in **d**, **h**, **l** and **m** (**e** and **i**).

(Supplementary Fig. 2). This range qualitatively establishes the size of our focus in all conditions.

We also identify the well-defined single-wavefront dislocations around the atomic defect by applying inverse FFT to the image with a size that is nearly identical, as shown in Fig. 2h (defect B). The robustness of the counted number of wavefront dislocations has been confirmed under various filter conditions (Supplementary Fig. 3), and similar characteristics are also observed in several other atomic defects utilizing the same tip (Supplementary Fig. 4). Obviously, the single-wavefront dislocations observed in this study deviates from the expected dual wavefront dislocations and cannot be only explained by the mechanism of pseudospin-related phase accumulation in real space. To investigate the origin of the single-wavefront dislocations, we conducted measurements of the spatial-resolved LDOS map and STS spectra for the representative atomic defect. Figure 2j displays the $dI/dV$ spatial map of defect B with a single wavefront dislocation. The LDOS surrounding the defect exhibits an approximately elliptical distribution, characterized by lower LDOS intensity in the upper left region, which suggests the presence of asymmetry in the system. The STS point spectra in Fig. 2k presents peaks (red arrows) with

approximately equidistant energy spacing of about 40 meV, which also shows asymmetric distribution around the defect at some energies. The origin of these peaks is attributed to the tip-induced quasibound states in graphene. Because of the different work functions between the tip and sample, the scanning probe can be regarded as a local top gate[38–40], generating a local potential to confine massless Dirac fermions in graphene. The range of the tip-induced potential at the gate voltage of 9 V, estimated based on the energy spacing of quasibound states[41–43], is approximately 15 nm. This potential range is on the same order of magnitude as the intervalley scattering region around the defect, characterized as a threefold $\sqrt{3} \times \sqrt{3} R 30°$ scattering pattern, which can potentially generate strong influence on the scattering processes. The existence of tip-induced gating effect is further corroborated by the observation of the charging peak[44], as shown in Fig. 2k (black arrow). The STS measurements under high magnetic fields reveal more pronounced asymmetric features for both quasibound states and the charging peak (Supplementary Fig. 5g).

In contrast, the distribution of LDOS in real space exhibits relative uniformity across the defect region containing two wavefront dislocations, indicating the absence of a rotational symmetry-breaking in

the system (Supplementary Fig. 5a–f). The source for the distinct potentials mostly arises from the different STM tips used in the measurements: one is a gold coated tungsten tip, introducing a finite electric field in graphene, and the other is a tungsten tip with the similar work function as graphene[44–47]. Moreover, the stability of the single-wavefront dislocations is preserved across different directions, as illustrated in Fig. 2l, m (Supplementary Figs. 6, 7 for wavefront dislocations at various energies and back-gate voltages). Based on the above observations, the anisotropic potential is the dominant factor causing discrepancy in the inverse FFT. The presence of a single wavefront dislocation can be attributed to the combined contribution of asymmetric potential induced inter-orbital angular momentum scatterings and pseudospin effects, which supports the hypothesis in Fig. 1 and will be further clarified in subsequent discussions.

## Theoretical analysis

To understand the emergence of the single-wavefront dislocations theoretically, we investigate the total LDOS modulation $\Delta\rho(\omega,\mathbf{r})$ generated by the intervalley scattering, which is measured in the experiment, by including both the defect-induced local potential and a rotationally asymmetric potential $V(r,\theta_\mathbf{r})$ spreading in a range around the defect ($r$ is the modulus of the vector $\mathbf{r}$). $\Delta\rho(\omega,\mathbf{r})$ can be formally written as the imaginary part of the complex scalar field (also see Supplementary Note 2):

$$\Delta\rho(\omega,\mathbf{r}) = \Delta\rho_A(\omega,\mathbf{r}) + \Delta\rho_B(\omega,\mathbf{r}) = -\frac{1}{\pi}Im\left\{e^{i\Delta\mathbf{K}\cdot\mathbf{r}}\left[a(r,\theta_\mathbf{r}) + b(r,\theta_\mathbf{r})e^{i2\theta_\mathbf{r}}\right]\right\}$$
$$= -\frac{1}{\pi}Im\left\{e^{i\Delta\mathbf{K}\cdot\mathbf{r}}\sum_{\Delta m}\left[a(r,\Delta m)e^{i\Delta m\theta_\mathbf{r}} + b(r,\Delta m)e^{i\Delta m\theta_\mathbf{r}}e^{i2\theta_\mathbf{r}}\right]\right\},$$
$$(1)$$

Here, $a(r,\Delta m)$ and $b(r,\Delta m)$ can be related to the amplitude of the LDOS modulation for the intervalley scattering on sublattice A and sublattice B ($\Delta\rho_A(\omega,\mathbf{r})$ and $\Delta\rho_B(\omega,\mathbf{r})$), respectively, with the transfer of the orbital angular momentum $\Delta m$. An additional term $e^{i2\theta_\mathbf{r}}$ appears along with $b(r,\Delta m)$, which corresponds to the contribution of pseudospin vector rotation. Moreover, a finite $\Delta m$ from the inter-orbital angular momentum scattering also introduces a phase $\Delta m\theta_\mathbf{r}$ on both sublattice components.

The situation of the wavefront dislocation depends on the phase singularity of this complex scalar field. When the potential $V(r,\theta_\mathbf{r})$ exhibits rotational symmetry, $\Delta m$ must be zero and the phase is $2\theta_\mathbf{r}$ on the B sublattice, which induces two wavefront dislocations in $\Delta\rho(\omega,\mathbf{r})$ (Supplementary Fig. 8). Note that the contribution from sublattice A to the overall LDOS modulation primarily affects the shape and position of the wavefront dislocations (see details in Supplementary Note 1). When the potential $V(r,\theta_\mathbf{r})$ shows rotational asymmetry, the inter-orbital angular momentum scatterings occur, and some terms can shift the phase to $\theta_\mathbf{r}$ rather than $2\theta_\mathbf{r}$. Under an appropriate potential $V(r,\theta_\mathbf{r})$, the low order terms $a(r,\Delta m=1)$ and $b(r,\Delta m=-1)$, both with phase $\theta_\mathbf{r}$, emerge as the dominant contributions near $\mathbf{r}=0$. They roughly contribute to the LDOS modulation as $\cos(\Delta\mathbf{K}\cdot\mathbf{r}+\theta_\mathbf{r})$ and a single wavefront dislocation is exhibited, as shown in Fig. 1f. In principle, the other terms may also contribute to the LDOS modulation, but the phase singularity which they induce could highly rely on the specific distributions of $a(r,\Delta m)$ and $b(r,\Delta m)$, and the wavefront dislocation can be far away from the defect. Thus, in this situation, $\Delta\rho(\omega,\mathbf{r})$ should exhibit one relatively robust additional dislocation near the defect.

We also performed the qualitatively calculated intervalley FFT-filtered $\Delta\rho(\omega,\mathbf{r})$. Besides a defect-induced delta potential, we also consider a Gaussian potential $-V_0e^{-r^2/R^2}$, and directly add one spread angle-dependent step potential field $V_1(r,\theta_\mathbf{r})$ on it to break the rotation symmetry (Supplementary Notes 3, 4)[48]. In Fig. 2i, we show the numerically calculated $\Delta\rho(\omega,\mathbf{r})$ for about 0.26 eV with

$\Delta\mathbf{K}=\left(-\frac{4\pi}{3\sqrt{3}a_{cc}},0\right)$, where $\Delta\mathbf{K}=\mathbf{K}-\mathbf{K}'$ is the momentum difference between two nearest valleys in the intervalley scattering process, and $a_{cc}=0.142$ nm is the length of carbon-carbon bond. Different from the result for the rotationally symmetric potential shown in Supplementary Fig. 8, only one wavefront dislocation is observed in the vicinity of the origin, and the orientation of this wavefront dislocation is still collinear to $\Delta\mathbf{K}$. It demonstrates that $2\pi$ rather than $4\pi$ phase is accumulated after a circulation around the defect, strongly suggesting that the phase $\theta_\mathbf{r}$ becomes dominant in $\Delta\rho(\omega,\mathbf{r})$ because of the interplay between the orbital angular momentum and pseudospin texture. The main features in Fig. 2i are well consistent with the experimental results in Fig. 2h, indicating the validity of our previous theoretical analysis. Particularly, the single wavefront dislocation exhibits a degree of robustness. Given that the pseudospin textures related to Dirac cones shown in Fig. 1c and the rotationally asymmetric potential near the defect shown in Fig. 1b persist over a range of energies, the scenario depicted in Fig. 1e, and the single wavefront dislocation could also hold for several energies (Supplementary Fig. 9). And considering the change of intervalley FFT-filtering direction only influences $\Delta\mathbf{K}$ but hardly changes the pseudospin vector rotation and the inter-orbital angular momentum scattering induced by a spread potential. The phase singularity, and the single wavefront dislocation in filtered LDOS modulation $\Delta\rho(\omega,\mathbf{r})$ should not change with different intervalley filtering directions (see Supplementary Fig. 10 and Supplementary Note 1).

## Wide existence of single-wavefront dislocations

The effect of rotational symmetry-breaking on inter-orbital angular momentum scatterings, which leads to the single-wavefront dislocations, is explicitly confirmed by introducing one-dimensional (1D) ripples to defect-rich graphene samples, as summarized in Fig. 3. Figure 3a exhibits a representative topographic image of 1D graphene ripples, showing a strained monolayer graphene with a periodicity of 15 nm[49]. The 1D strained system exhibits a strong rotational symmetry-breaking, as observed in both its structure and LDOS (Supplementary Fig. 11). Here, a tungsten tip is utilized to capture the LDOS in distinct regions, revealing the presence of a flat band corresponding to the pseudo-Landau level, along with pseudomagnetic confinement states situated in between, which is consistent with the previous experiments[49,50].

In Fig. 3b, a prominent threefold scattering feature is observed for the defect (defect C), which is located within the periodically strained structure with confinement states. We employ FFT analysis to visualize the intervalley scattering in Fig. 3c. The resulting FFT yields six distinct inner peaks with pronounced asymmetric intensities, hinting that the presence of the asymmetric structure and potential in this system. Similarly, we perform inverse FFT for intervalley scattering signals to visualize the LDOS modulation in real space. The single-wavefront dislocations around the defect is robust at various orientations, as depicted in Fig. 3d–f (Supplementary Fig. 11 for wavefront dislocations at different energies). The study reveals that the anisotropic substrate, along with its corresponding asymmetric electromagnetic potential, can also induce inter-orbital angular momentum scatterings. The orbital angular momentum-modulated phase singularity consequently results in the phase change from $2\theta_\mathbf{r}$ to $\theta_\mathbf{r}$ in real space. This substrate engineering provides a versatile platform for creating customized structures and shows the potential to manipulate intervalley scattering processes in future experiments.

In conclusion, our study uncovers the crucial role of orbital angular momentum in processes of Friedel oscillations. The breakdown of orbital angular momentum conservation and the occurrence of inter-orbital angular momentum scatterings lead to additional phase singularities, and result in the transformation of double Y-shaped fringes into a single wavefront dislocation. Through our numerical simulations, we successfully capture the orbital-structured pseudospin and the anomalous single-wavefront dislocations in real

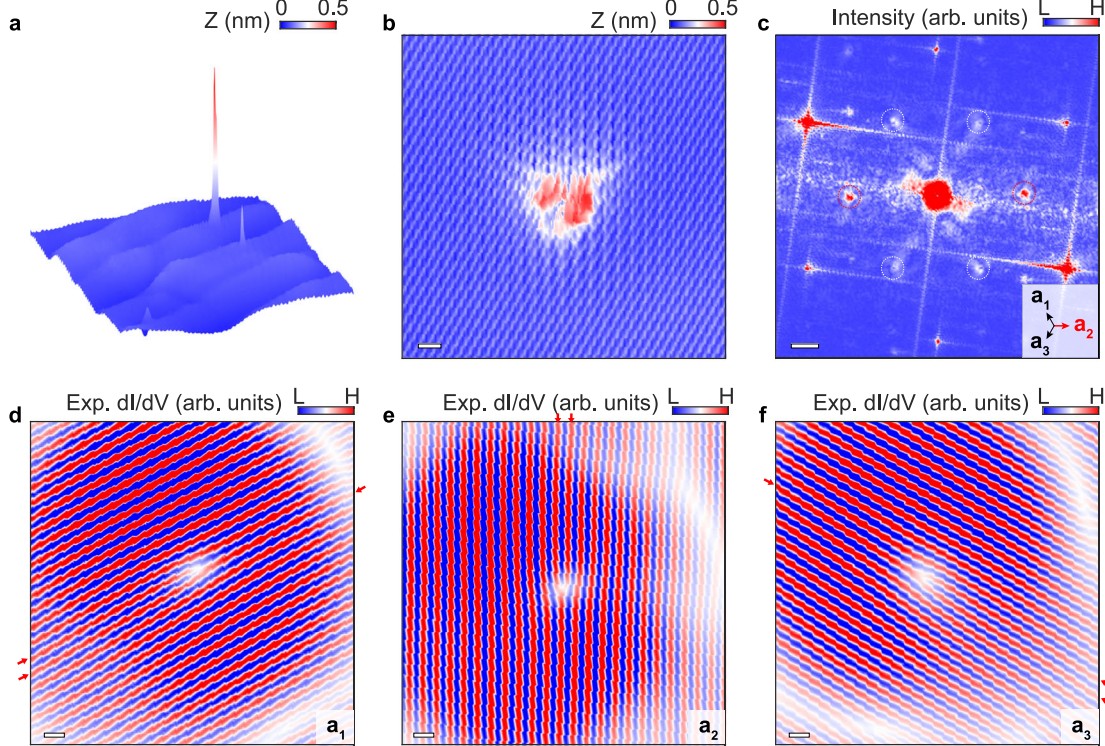

**Fig. 3 | Single-wavefront dislocations induced by the anisotropic potential.**
**a** STM geometry of the periodic strained graphene with a defect on it. The size of the scanning window is $70 \times 70$ nm$^2$. **b** An enlarged topographic image of a defect obtained through the tungsten tip (defect C, $V_b = 0.4$ V) on the strained structure. Scale bar: 0.5 nm. **c** FFT image of the defect in panel **b**. The inner asymmetric points result from intervalley scattering processes. The directions of intervalley scattering are marked in the inset with $a_1$, $a_2$, and $a_3$. Scale bar: 5 nm$^{-1}$. **d–f** Experimental wavefront dislocations in real space which are filtered along $a_1$, $a_2$, and $a_3$ directions, which is perpendicular to the intervalley scattering peaks in panel **c**. Scale bar: 0.5 nm. The wavelength is about 0.37 nm. The change of wavefronts passing through the defects are indicated by the red arrows in **d–f**.

space. Our study presents an innovative approach to deciphering the orbital angular momentum effect by examining phase shifts and wavefront dislocations at the atomic scale. Looking ahead, the successful implementation of optical and electron beams with orbital angular momentum has demonstrated promising potential across various fields[17–23]. Our results on Dirac fermions with orbital angular momentum advances the understanding of phase singularities and wavefront dislocations in lower dimensions within condensed matter physics. The concept of the inter-orbital angular momentum scattering induced phase singularity can be extended to arrays or a larger scale, which may offer potential applications in nanodevices, electron-optical setups[38], novel microscopy techniques[23], and quantum communication systems[20,21].

## Methods
### Sample fabrication
To obtain graphene samples with a high number of hydrogen chemisorption defects, we utilized two methods to intentionally introduce these defects. In the first method, we fabricated graphene samples using low-pressure chemical vapor deposition (LPCVD). We introduce hydrogen defects by adjusting the hydrogen ratio during the growth process. Subsequently, we transferred graphene to hBN substrates to investigate gate-dependent states in the vicinity of the defects. The contacts were made by evaporating Ti/Au (3 nm/30 nm). The samples were annealed at 300 °C in LPCVD and ultra-high vacuum before the STM measurements. The second approach involves directly creating graphene defects through the STM tip pulses on the Ge substrate. This method relies on the relatively low Ge-H bond energy and the desorption of hydrogen atoms facilitated by the pulse stimulation.

### STM/STS measurements
STM and STS measurements were performed with commercial STM systems from UNISOKU (USM-1300/1400/1500) in an ultrahigh vacuum chamber with a pressure of about $10^{-11}$ Torr. The topography images were taken by turning on the feedback circuit. The $dI/dV$ spectra were taken with a standard lock-in technique by turning off the feedback circuit and using a 793 Hz, 1–5 mV$_{pp}$ modulation added to $V_b$ with the time constant of 3–30 ms. $d^2I/dV^2$ was obtained through numerical differentiation. The STM tips were obtained by etching tungsten wires, and they were cleaned by e-beam heating before the measurements. The work functions of graphene sample and STM tips vary between 4.5–4.7 eV for graphene, 3.6–4.0 eV for gold-coated tips, and 4.86 eV for Pt-Ir tips[47], 4.5 eV for tungsten tips[51,52], respectively. The experimental results shown in the main text were all taken at around 4.2 K.

## Data availability
The data that support the findings of this study are available from the corresponding author upon request.

## Code availability
The code that supports the findings of this study is available from the corresponding author upon request.

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

## Acknowledgements

We thank H. Zhou, Z. Zhan, Y. Su, W. Zhi, and P. Wu for discussions. Q.S. acknowledges the Strategic priority Research Program of Chinese Academy of Sciences (Grant No. XDB28000000). The research done by Y.L., Y.R. and L.H. was supported by the National Key R and D Program of

China (Grant No. 2021YFA1401900). L.H. acknowledges the National Key R and D Program of China (Grant No. 2021YFA1400100), the National Natural Science Foundation of China (Grants No. 12141401), and the Fundamental Research Funds for the Central Universities. The research done by Y.Z. and Q.S. was supported by the National Natural Science Foundation of China (Grant No. 12374034, and No. 11921005), and the Innovation Program for Quantum Science and Technology (2021ZD0302403). Y.R. acknowledges the China Postdoctoral Science Foundation (2023M740296).

## Author contributions

Y.L. and L.H. conceived the work and designed the research strategy. Y.L. fabricated the samples and carried out the STM and STS measurements. Y.R., C.Y., X.Z. and Q.Y. participated in the measurements and provided experimental data. Y.L. and Y.Z. performed the data analysis under the supervision of L.H. and Q.S. Y.Z. carried out the theoretical analysis and Green's function calculations under the supervision of Q.S. Y.L. and Y.Z. wrote the paper with input from L.H. and Q.S.

## Competing interests

The authors declare no competing interests
