## [Peer Review File · Nature Communications]

Visualizing a single wavefront dislocation induced by orbital angular momentum in grapheneREVIEWER COMMENTS

Reviewer #1 (Remarks to the Author):

The authors have studied the role of orbital angular momentum in phase singularities in graphene, particularly through scanning tunneling microscopy and spectroscopy. The experimental results show the appearance of single wave front dislocations in the interference pattern. This is explained by the scattering between different orbital angular momentum states in the presence of a rotationally asymmetric potentials which result in a transition of the interference pattern from two wave front dislocations (observed in reference 9 for a symmetric potential) to one wave front dislocation. This work provides the first local imaging of the effect of orbital angular momentum on wave front dislocations at the nanoscale and can be important for future local studies of other systems. The experimental results are of high quality and the interpretations are reasonable and supported well by theory. I recommend the paper for publication. I only have also the following optional suggestions/questions:

1- I suggest authors to describe the wave front images in a more transparent way to make the paper more accessible to general readers. For example, I wonder if the color scale in wave fronts and potential are related in Fig 1a and 1b? It might be better to also use different color scales in Fig 1c vs 1d or Fig 1e vs Fig 1f. Furthermore, I suggest showing the total calculated charge density modulations in Fig. 1d instead of only sublattice B. This allows better comparison between measured data and theory and avoid confusion as to relate to previous studies.

2- The figure labels in Fig 2 are very small and hard to see.

3- The authors provide evidence for the presence of an asymmetric potential by either an approximately elliptical distribution of LDOS states around the defect, or asymmetric intensity in FFT peaks. The asymmetric intensity in FFT peaks is also observable in data in reference 9 which claim the presence of a rotationally symmetric potential. Can the authors comment on that?

Reviewer #2 (Remarks to the Author):

This paper has studied the role of orbital angular momentum (OAM) in wavefront dislocations (WfDs) in graphene, employing scanning tunneling microscopy and spectroscopy. They noticed that when an incident wave from sublattice B in K valley is scattered by an isotropic defect-induced local potential at sublattice A, it will get back to sublattice B in K' valley with no change in the OAM of electrons but $-2\theta_q$ rotation in the pseudospin vector due to the distinctive pseudospin textures for two valleys. And $2\theta_q$ corresponds to a spatial phase $2\theta_r$ in the local density of states (LDOS) modulation through the Fourier transformation and brings 2 WfDs. On the other hand, if an anisotropic potential is applied, the OAM of electrons will be changed, resulting inter-OAM scatterings. In this case, the interference between the incident wave and reflected wave brings additional θ_r , making the phase shift in LDOS nodulation for B sublattice from $2\theta_r$ to θ_r . When they explored the inter-OAM scattering on their graphene device under STM, they observed two distinctive defects (Defect A&B). The inverse FFT of defect A shows two Y-fringes, indicating two WfDs around defect A. Defect A's experimental data is in agreement with the

calculation result counting only a defect-induced delta potential, which further illuminates the $2\theta_r$ situation that pseudospin vector rotations of $2\theta_q$ by intervalley scatterings brings 2 WfDs. However, the inverse FFT of defect B shows only one Y-fringe, corresponding to single WfDs. They attributed this anomaly to the tip-induced highly asymmetric potential by showing the elliptical distribution of LDOS, which implies an asymmetric potential distribution around the defect, and the tip-induced equidistant quasibound states and a charging peak in derivative STS spectra. They believe that different STM tips, owning different tip-induced gating effect, are the source for the distinct arising potentials which decide the number of WfDs. The emergence of such single WfD is preserved across different directions. Furthermore, they found that, similar to the tip-induced asymmetric potential, the anisotropic potential from the substrate can also induce inter-OAM scatterings and single WfD, which is preserved across different directions as well.

Overall, I find this experimental observation interesting and have not come across any previous reports on this topic. However, I would appreciate it if the authors could address the following questions:

1. What is the dominant factor causing the difference between the inverse FFT figures of defects A and B? If the difference is due to intrinsic variations between these defects, what specifically accounts for this dissimilarity? Alternatively, if the discrepancy arises from using different tips during measurement, it should be clearly indicated in each figure which type of tip was employed."
2. Regarding the sentence in line 139-141, it appears incomplete and unclear. Consider revising it for coherence and completeness. For example: "The sentence 'Obviously, the observed anomalous WfD here, beyond the anticipation derived from pseudospin-induced phase accumulation' appears incomplete and lacks clarity. Please clarify or provide additional context to enhance understanding."
3. In lines 152-154, it mentions that "the range of the tip-induced potential at the gate voltage of 9 V ... is approximately 15 nm, which corresponds to the dimensions of the scattering region encompassing the defect." It's unclear why 15 nm is emphasized here. Is it related to the later-mentioned periodicity of a strained monolayer graphene (line 223) or to the 3.7 Å-wavelength LDOS modulations (line 126) and a 15-wavelength distance (line 133)? If it's the latter, the calculation should be clarified, as there seems to be a discrepancy in the explanation.
4. The mirror phenomena in the two panels of Fig. 2e are mentioned but not explained. Consider providing an explanation for these mirror features, including their robustness in different energies and directions.
5. A minor suggestion is to enhance the clarity of Fig. 2g by showing the cut-off line (x-axis) in a larger scope picture, making it easier for readers to understand where this figure is acquired from.

Reviewer #3 (Remarks to the Author):

Report on manuscript NCOMMS-24-02147-T.

This paper describes measurement and theory of wavefront dislocations observed with atomic defects in graphene. It builds and advances the previous work in Reference 9, which showed that the Berry phase properties in graphene result in a wavefront dislocation when examining the intervalley scattering in the STM images. This manuscript expands the previous work to include aspects of orbital angular momentum and how it changes the number of wavefront dislocations in the defect scattering images. The authors demonstrate the required symmetry breaking needed for the orbital effects with both the potential from the STM probe tip and from also inherent asymmetric potentials in graphene. This work will be important in future studies of moire materials also in ABC stacked graphene, where orbital effects are of interest. I recommend publication in Nature Communications, with some optional suggestions for the authors to consider.

Some optional suggestions the authors can consider to improvement of the manuscript.

1. 99.9 percent of the readers are not familiar with the wavefront images as in Fig. 1d etc.. The description from lines 49 to 72 is not that clear or transparent, and also how the inverse FFT from selecting two peaks in the FFT image give this phase image. I think a little more description here would be useful for the general reader.

2. I encourage the authors to print the manuscript to see how small some text and images are. While figures can be blow up on a screen, many details cannot be seen in print copies. For example, the FFT insets in Fig. 2b and d are too small. I recommend to make these the same size as the images and just expand the figure.

3. The text labels in Fig. 1d and f cannot be seen in a print copy.

4. I strongly recommend not to use the acronym IVS, WFD, and OAM. It makes reading the manuscript difficult and unpleasant. It is much better to just spell these out when used, no matter how many times they appear. The use of acronyms should be only used for very common terms, like STM, etc.

Reviewer #1 (Remarks to the Author):

The authors have studied the role of orbital angular momentum in phase singularities in graphene, particularly through scanning tunneling microscopy and spectroscopy. The experimental results show the appearance of single wave front dislocations in the interference pattern. This is explained by the scattering between different orbital angular momentum states in the presence of a rotationally asymmetric potentials which result in a transition of the interference pattern from two wave front dislocations (observed in reference 9 for a symmetric potential) to one wave front dislocation. This work provides the first local imaging of the effect of orbital angular momentum on wave front dislocations at the nanoscale and can be important for future local studies of other systems. The experimental results are of high quality and the interpretations are reasonable and supported well by theory. I recommend the paper for publication. I only have also the following optional suggestions/questions:

Reply: We are glad that referee #1 recommended our manuscript to be published. We appreciate the comments of the referee, and we revise our manuscript point by point according to the suggestions/questions of Reviewer #1. The revised sections are marked with red font and red underline in the "Revised manuscript with highlighted modifications" and "Revised Supplementary Information with highlighted modifications."

1. I suggest authors to describe the wave front images in a more transparent way to make the paper more accessible to general readers. For example, I wonder if the color scale in wave fronts and potential are related in Fig 1a and 1b? It might be better to also use different color scales in Fig 1c vs 1d or Fig 1e vs Fig 1f. Furthermore, I suggest showing the total calculated charge density modulations in Fig. 1d instead of only sublattice B. This allows better comparison between measured data and theory and avoid confusion as to relate to previous studies.

Reply: We thank the referee for these helpful comments and suggestions.

1. A relationship exists between the color scale in wave fronts and potential, as they both serve as schematic diagrams that depict the magnitude of intensity. In the case of the wavefront, color is utilized to represent the intensity of the wave, with red indicating high intensity and blue indicating low intensity. When $m=0$, it is schematically assumed that the wave intensity is concentrated at the center. Conversely, when $m=1$, the presence of a phase singularity at the center of the wave results in the disappearance of intensity at that point, leading to a donut-shaped distribution. For potential fields, red is indicative of high potential fields, while blue signifies low potential fields. In the revised Fig. 1a and 1b, we have included a text label near the color bar to explicitly indicate that the color scale represents both wavefront intensity and potential intensity. Furthermore, we have revised the figure captions to

clarify them (line 352 highlighted by red color).

2. In the new version manuscript, we have employed distinct color scales to differentiate between the phase distribution and intensity distribution. Furthermore, we have emphasized the difference between the phase and amplitude color scales in the figure caption, stating that the color gradient from black to red signifies the variation of the phase (line 356 highlighted by red color).

3. According to the suggestion of the reviewer, in the revised manuscript, we first put the original Fig. 1d which schematically shows the LDOS modulation on sublattice B as supplementary figure Fig. S1. In the new Fig. 1d (also the Fig. R1 in this response letter), we schematically show the total charge density modulation $\Delta\rho(\Delta\vec{K}, \vec{r}) = \Delta\rho_A(\Delta\vec{K}, \vec{r}) + \Delta\rho_B(\Delta\vec{K}, \vec{r})$ on both sublattices A and B, where the variations of amplitudes with the radial direction are also included. Clearly, this new Fig. 1d which exhibits double Y-shaped fringes is very consistent with measured and calculated data in Fig. 2d and Fig. 2e, allowing better comparison. We also added some sentences on page 3 in the main text (highlighted by red color in lines 70-74) and supplementary information (highlighted by red color in lines 87-89, 92, 101-108 and Fig. S1) to explain the details of this new Fig. 1d.

Fig. R1. A schematic diagram to show wavefront dislocations in total LDOS modulation $\Delta\rho(\Delta\vec{K}, \vec{r}) = \Delta\rho_A(\Delta\vec{K}, \vec{r}) + \Delta\rho_B(\Delta\vec{K}, \vec{r})$ with double Y-shaped fringes. The white disk depicts the defects.

2. The figure labels in Fig 2 are very small and hard to see.

Reply: We would like to express our gratitude to the referee for this comment. We have enlarged the font size for all figures

3. The authors provide evidence for the presence of an asymmetric potential by either an approximately elliptical distribution of LDOS states around the defect, or asymmetric intensity in FFT peaks. The asymmetric intensity in FFT peaks is also observable in data in reference 9 which claim the presence of a rotationally symmetric potential. Can the authors comment on that?

Reply: We greatly appreciate the referee for the question.

In this work, the asymmetric FFT signal serves as an auxiliary method to hint at the presence of rotational symmetry-breaking in our system, but it does not directly illustrate the existence of rotational symmetry-breaking potential field. In Figure 3, both its structure and LDOS break the rotational invariance of the system. Therefore, the asymmetric FFT observed in Figure 3c can be attributed to the anisotropic strain induced lattice distortion or anisotropic electromagnetic potential.

In most cases of STM measurements, the tip usually generates a relatively smooth potential field, and its impact on inter-valley scattering is weak. The FFT signal may not exhibit noticeable asymmetry. Moreover, even if the rotational symmetry of the system is broken, a certain level of strength is required to generate a single wavefront dislocation. Minor deviations from rotational symmetry are unlikely to significantly affect the coupling of angular momentum in scattering processes. Therefore, although the weak asymmetric intensity in the FFT peaks is observed in Reference 9, the anisotropic potential field can be assumed not strong enough, so that double wavefront dislocations are observed.

We appreciate the referee's comment on this matter. As a result, we have revised the manuscript to provide a more accurate description of the intensity of FFT peaks (lines 221-234 highlighted by red color).

Reviewer #2 (Remarks to the Author):

This paper has studied the role of orbital angular momentum (OAM) in wavefront dislocations (WfDs) in graphene, employing scanning tunneling microscopy and spectroscopy. They noticed that when an incident wave from sublattice B in K valley is scattered by an isotropic defect-induced local potential at sublattice A, it will get back to sublattice B in K' valley with no change in the OAM of electrons but $-2\theta_q$ rotation in the pseudospin vector due to the distinctive pseudospin textures for two valleys. And $2\theta_q$ corresponds to a spatial phase $2\theta_r$ in the local density of states (LDOS) modulation through the Fourier transformation and brings 2 WfDs. On the other hand, if an anisotropic potential is applied, the OAM of electrons will be changed, resulting inter-OAM scatterings. In this case, the interference between the incident wave and reflected wave brings additional θ_r , making the phase shift in LDOS modulation for B sublattice from $2\theta_r$ to θ_r . When they explored the inter-OAM scattering on their graphene device under STM, they observed two distinctive defects (Defect A&B). The inverse FFT of defect A shows two Y-fringes, indicating two WfDs around defect A. Defect A's experimental data is in agreement with the calculation result counting only a defect-induced delta potential, which further illuminates the $2\theta_r$ situation that pseudospin vector rotations of $2\theta_q$ by intervalley scatterings brings 2 WfDs. However, the inverse FFT of defect B shows only one Y-fringe, corresponding to single WfDs. They attributed this anomaly to the tip-induced highly asymmetric potential by showing the elliptical distribution of LDOS, which implies an asymmetric potential distribution around the defect, and the tip-induced equidistant quasibound states and a charging peak in derivative STS spectra. They believe that different STM tips, owning different tip-induced gating effect, are the source for the distinct arising potentials which decide the number of WfDs. The emergence of such single WfD is preserved across different directions. Furthermore, they found that, similar to the tip-induced asymmetric potential, the anisotropic potential from the substrate can also induce inter-OAM scatterings and single WfD, which is preserved across different directions as well.

Overall, I find this experimental observation interesting and have not come across any previous reports on this topic. However, I would appreciate it if the authors could address the following questions:

Reply: We are grateful for the interest shown by Referee #2 in our observation. We will address the questions raised by Reviewer #2 point-by-point. The revised parts are highlighted by blue font and curved underline in the "Revised manuscript with highlighted modifications" and "Revised Supplementary Information with highlighted modifications."

1. What is the dominant factor causing the difference between the inverse FFT figures

of defects A and B? If the difference is due to intrinsic variations between these defects, what specifically accounts for this dissimilarity? Alternatively, if the discrepancy arises from using different tips during measurement, it should be clearly indicated in each figure which type of tip was employed."

Reply: Thanks to the referee for this comment.

The primary factor causing the difference in the inverse FFT is the presence of a strong asymmetric potential, or more specifically, the asymmetric potential-induced inter-angular momentum scatterings. A single wavefront dislocation occurs when a strong asymmetric potential is present, while a dual wavefront dislocation occurs without a strong asymmetric potential. As defects A and B are both classified as hydrogen chemisorption defects, the difference does not arise from intrinsic variations between these defects.

Figure 2 demonstrates that the differences in potential, which cause the discrepancy in inverse FFT, are attributed to the distinct work functions between different tips and graphene. A greater disparity in work function, such as with an Au-coated tip, increases the likelihood of inducing local potential or generating an asymmetric potential. Simultaneously, this can result in the formation of a single wavefront dislocation through the coupling of the orbital angular momentum of electrons.

In response to the helpful suggestion from Reviewer #2, we have highlighted the primary factor responsible for the disparity between the inverse FFT figures (lines 165-166), the types of STM tips in the caption of each figure in the revised manuscript (lines 371, 378 and 390 highlighted by blue color).

2. Regarding the sentence in line 139-141, it appears incomplete and unclear. Consider revising it for coherence and completeness. For example: "The sentence 'Obviously, the observed anomalous WfD here, beyond the anticipation derived from pseudospin-induced phase accumulation' appears incomplete and lacks clarity. Please clarify or provide additional context to enhance understanding."

Reply: We thank the referee's comment on this. We complete this sentence as follows: Obviously, the single wavefront dislocation observed in this study deviates from the expected dual wavefront dislocations and cannot be only explained by the mechanism of pseudospin-related phase accumulation in real space (see lines 140-142 highlighted by blue color).

3. In lines 152-154, it mentions that "the range of the tip-induced potential at the gate voltage of 9 V ... is approximately 15 nm, which corresponds to the dimensions of the scattering region encompassing the defect." It's unclear why 15 nm is emphasized here. Is it related to the later-mentioned periodicity of a strained monolayer graphene (line

223) or to the 3.7 Å-wavelength LDOS modulations (line 126) and a 15-wavelength distance (line 133)? If it's the latter, the calculation should be clarified, as there seems to be a discrepancy in the explanation.

Reply: We are grateful to the referee for requesting a clearer description of the 15 nm range.

We emphasize the 15 nm range (potential range) here because the intervalley scattering around the defect, characterized as the threefold $\sqrt{3}\times\sqrt{3}$ scattering pattern, occurs within approximately 10 nm – 15 nm (scattering range). Given that potential and scattering ranges are of the same order of magnitude, the rotational symmetry-breaking potential within this 15 nm range can significantly influence the scattering processes, which is an essential factor in generating the single wavefront dislocation. The presence of a single wavefront dislocation requires a strong asymmetric potential. If the range of potential is large, such as 200 nm, the potential changes could be very slow near nanoscale defects, which may have little effect on scattering processes.

Therefore, this emphasis is not related to the periodicity of the strained monolayer graphene or the 15-wavelength distance mentioned in the text. It relates to the strong asymmetric potential around the defect. We have addressed this in the revised manuscript to make it clearer (See lines 152-155 highlighted with blue color).

4. The mirror phenomena in the two panels of Fig. 2e are mentioned but not explained. Consider providing an explanation for these mirror features, including their robustness in different energies and directions.

Reply: We are grateful to the referee for requesting an improved explanation.

Here the mirror phenomenon means the numerical simulated results are very similar to experiment observations. The similarities can be mainly summarized into three aspects.

1. Firstly, only one wavefront dislocation appears near the defect, as shown in new Fig. 2h and Fig. 2i (i.e., Fig.2e in the old version manuscript). It indicates that θ_r becomes dominant contributions near the center, due to the interplay between inter-orbital angular momentum coupling and intervalley scattering induced pseudospin vector rotation, which is also reflected by the formula analysis in Eq. (1) in the main text.

2. Secondly, the single wavefront dislocation is robust as the energy varies, as shown in Extended Data Fig. 8. This is reasonable, given that the pseudospin textures related to Dirac cones shown in Fig. 1c, and the rotationally asymmetric potential near the defect shown in Fig. 1b both persist over a range of energies. Thus, for electrons in different energies, the picture of intervalley scattering and inter-orbital angular momentum coupling shown in Fig. 1e should still work.

3. Thirdly, the single wavefront dislocation is observed in different inverse FFT directions, as shown in Figs. 2l,m and Extended Data Fig. 9. This can be understood by Eq. (1) in the main text and Eq. (31) in the supplementary information. In our system, the change of intervalley FFT-filtering direction only influences $\Delta\vec{k}$ but hardly alters the pseudospin vector rotation and inter-orbital angular momentum coupling. Therefore, the phase singularity and the single wavefront dislocation in filtered LDOS modulation $\Delta\rho(\omega, \vec{r})$ should remain unchanged regardless of the intervalley filtering direction.

In the revised manuscript, we added several sentences in the first paragraph on page 7 to explain the mirror features in Figs. 2h,i and includes the robustness in different energies and directions as well (see lines 207-216 highlighted with blue color).

5. A minor suggestion is to enhance the clarity of Fig. 2g by showing the cut-off line (x-axis) in a larger scope picture, making it easier for readers to understand where this figure is acquired from.

Reply: We appreciate the referee's suggestion.

We have presented a linecut in real space in Figure 2j (i.e., Fig.2g in the old version of manuscript) to indicate the origin of the STS. As the linecut is not sufficiently long, we have emphasized the length of the spectra in the figure caption (see lines 382-383 highlighted in blue color).

Reviewer #3 (Remarks to the Author):

Report on manuscript NCOMMS-24-02147-T.

This paper describes measurement and theory of wavefront dislocations observed with atomic defects in graphene. It builds and advances the previous work in Reference 9, which showed that the Berry phase properties in graphene result in a wavefront dislocation when examining the intervalley scattering in the STM images. This manuscript expands the previous work to include aspects of orbital angular momentum and how it changes the number of wavefront dislocations in the defect scattering images. The authors demonstrate the required symmetry breaking needed for the orbital effects with both the potential from the STM probe tip and from also inherent asymmetric potentials in graphene. This work will be important in future studies of moire materials also in ABC stacked graphene, where orbital effects are of interest. I recommend publication in Nature Communications, with some optional suggestions for the authors to consider.

Some optional suggestions the authors can consider to improvement of the manuscript.

Reply: We are grateful that Reviewer #3 recommended the publication and gave us helpful suggestions. We have carefully revised our manuscript, addressing each of the referee's suggestions. The revised parts are highlighted by purple font and straight underline in "Revised manuscript with highlighted modifications" and "Revised Supplementary Information with highlighted modifications."

1. 99.9 percent of the readers are not familiar with the wavefront images as in Fig. 1d etc.. The description from lines 49 to 72 is not that clear or transparent, and also how the inverse FFT from selecting two peaks in the FFT image give this phase image. I think a little more description here would be useful for the general reader.

As introduced in Figs. 1a,c,d, the pseudospin vector rotation induced by IVS leads to 4π phase accumulation encircling the defect and, consequently, generates two WfDs around the defect^{9,31,32}

Reply: We appreciate the referee reminding us to add more descriptions for readers. To make it clear from the beginning, we added more descriptions to interpret the wavefront images, and why the phase is related to FFT peaks in the revised manuscript and figure captions (Please see lines 27, 42-44, 51-80, 121-128 in the revised main text, and 360-361 in the figure caption. They are highlighted by purple color).

2. I encourage the authors to print the manuscript to see how small some text and images are. While figures can be blow up on a screen, many details cannot be seen in print copies. For example, the FFT insets in Fig. 2b and d are too small. I recommend to make these the same size as the images and just expand the figure.

Reply: We would like to express our gratitude for the referee's valuable comments. We have enlarged the inset, presenting them as isolated figures as Fig. 2c and Fig. 2g. Additionally, we have enlarged the font size for all figures as well.

3. The text labels in Fig. 1d and f cannot be seen in a print copy.

Reply: We appreciate the referee's valuable comments.

We have adjusted them to ensure that the text labels are good enough to see. Moreover, we have made the font bold for Fig. 1d and 1f.

4. I strongly recommend not to use the acronym IVS, WFD, and OAM. It makes reading the manuscript difficult and unpleasant. It is much better to just spell these out when used, no matter how many times they appear. The use of acronyms should be only used for very common terms, like STM, etc.

Reply: We agree it is unpleasant for the reader to read these words. Therefore, we replaced IVS, WFD, and OAM with their respective full forms - intervalley scattering, wavefront dislocation, and orbital angular momentum - in both the revised manuscript and supplementary materials.

REVIEWERS' COMMENTS

Reviewer #2 (Remarks to the Author):

The authors have addressed my comments and I support its publication ASIS.